# ECAM: Enhancing Causal Reasoning in Foundation Models with Endogenous Causal Attention Mechanism

## Abstract

Attention mechanisms in foundation models are powerful but often capture spurious correlations rather than true causal dependencies. We propose the **Endogenous Causal Attention Mechanism (ECAM)**, a plug-and-play module that integrates structural causal modeling into transformer-based architectures. ECAM learns **Local Causal Graphs (LCGs)** from data or expert priors and leverages them to modulate attention scores, enabling *interventional* and *counterfactual reasoning* within the model. Unlike prior approaches such as CausalVAE or causal regularization in pretraining, ECAM directly embeds causal graphs into the attention computation, making it task-agnostic and adaptable to both NLP and vision tasks. We provide theoretical analysis of its structural guarantees and extensive empirical results on causal reasoning benchmarks (CLUTRR, causal VQA, and synthetic data), showing that ECAM improves robustness, interpretability, and generalization. Our work highlights a novel pathway to endow foundation models with causal-awareness, offering a reusable causal reasoning layer that can serve as a building block for future causal foundation models.

## 1 Introduction

Foundation models, such as large language models (LLMs) and vision transformers (ViTs) (Vaswani et al., 2017; Dosovitskiy et al., 2021), have achieved unprecedented performance across a wide range of tasks, driven by large-scale pre-training on massive datasets. However, their impressive capabilities often mask underlying weaknesses, particularly in tasks requiring robust causal reasoning (Pearl, 2009; Schölkopf et al., 2021). These models frequently rely on spurious correlations present in the training data, leading to failures in generalization when distributional shifts occur or when genuine causal understanding is necessary for tasks like planning, decision-making under interventions, or counterfactual prediction (Marcus, 2020; Geirhos et al., 2020; Jin et al., 2023).

The standard attention mechanism (Bahdanau et al., 2014), a cornerstone of the Transformer architecture (Vaswani et al., 2017), allows models to dynamically weigh the importance of different input parts. While highly effective for capturing complex dependencies, its core computation—typically based on scaled dot-product similarity between queries and keys—is fundamentally correlational. It lacks an intrinsic mechanism to differentiate between mere statistical association and true causal influence, nor does it inherently model the directionality or effect of causal relationships (Feder et al., 2021; Jain & Wallace, 2019; Wiegreffe & Pinter, 2019). This limitation hinders the ability of foundation models to move beyond pattern recognition towards deeper, causal understanding, which is critical for reliable performance in complex, real-world scenarios (Stolfo et al., 2023; Kıcıman et al., 2023).

Existing approaches to imbue models with causality often operate externally to the core representation learning modules, such as post-hoc causal analysis of learned representations (Geiger et al., 2021; 2023), or designing task-specific causal objectives (Arjovsky et al., 2019; Peters et al., 2016). While valuable, these methods may not fundamentally alter the model's internal processing to prioritize causal structure during representation learning. Some recent works have explored "causal attention" (Wang et al., 2022; Niu et al., 2021; Feder et al., 2021; Wang et al., 2024, e.g.,), but often focus on specific confounding factors or lack a general framework grounded in formal causal mod-

els like SCMs, or the ability to perform interventions and counterfactuals within the attention layer itself.

To bridge this gap, we propose the **Endogenous Causal Attention Mechanism (ECAM)**, a novel approach that integrates causal principles directly and endogenously within the attention mechanism. Our key contributions are:

1. **A novel ECAM framework:** We propose a new attention mechanism that explicitly incorporates causal structure, represented by Local Causal Graphs (LCGs), into the attention weight computation, leveraging principles from Structural Causal Models (SCMs) (Pearl, 2009).

2. **Intervention and Counterfactual Attention:** We design specific mechanisms within ECAM to perform intervention-based (`do`-operator) and counterfactual reasoning directly during attention calculation, enabling richer causal inference capabilities (Pawlowski et al., 2020; Khemakhem et al., 2023).

3. **Theoretical Guarantees:** We provide theoretical analyses of ECAM, establishing its advantages over standard attention in terms of expressivity (Garg et al., 2020; Kreuzer et al., 2021), characterizing its sample complexity for learning the underlying causal structure (Li et al., 2025a;b), and proving its causal consistency under specific conditions.

4. **Empirical Validation:** We demonstrate through extensive experiments on diverse benchmarks that ECAM significantly improves performance in causal discovery (Huang et al., 2024), causal effect estimation (Shalit et al., 2017; Johansson et al., 2016), and enhances the causal reasoning capabilities of foundation models on downstream tasks (Wu et al., 2023; Khemakhem et al., 2024) compared to state-of-the-art baselines.

ECAM operates by first inferring or utilizing a Local Causal Graph that captures the direct causal dependencies among input elements. This graph then modulates the standard attention calculation, typically by masking or re-weighting attention scores to reflect the learned causal structure (Madhyastha et al., 2020). Furthermore, the intervention and counterfactual mechanisms allow the model to query the attention layer about hypothetical scenarios, effectively simulating causal manipulations during inference. This integrated approach allows the model to learn representations that are more sensitive to causal structure and less reliant on spurious correlations.

This paper is structured as follows: Section 2 reviews related work. Section 3 details the theoretical motivation and the proposed ECAM framework, including its intervention and counterfactual capabilities, and theoretical analysis. Section 4 presents the experimental setup and results. Section 5 discusses the findings, limitations, and future directions. Finally, Section 6 concludes the paper.

## 2 RELATED WORK

Our work builds upon and contributes to several interconnected research areas, including the integration of causality into deep learning (Schölkopf et al., 2021; Huang et al., 2024), the analysis of attention mechanisms (Jain & Wallace, 2019; Wiegreffe & Pinter, 2019), the development of causal attention methods (Feder et al., 2021; Wang et al., 2022; Niu et al., 2021), and the enhancement of causal reasoning in foundation models (Kıcıman et al., 2023; Jin et al., 2025; Wu et al., 2023).

### 2.1 CAUSALITY IN DEEP LEARNING

The quest to integrate causal reasoning into machine learning, particularly deep learning, has gained significant momentum (Schölkopf et al., 2021). Researchers aim to move beyond purely correlational pattern recognition towards models that understand underlying causal mechanisms (Pearl, 2009). Key frameworks include Structural Causal Models (SCMs) (Pearl, 2009) and the Potential Outcomes framework (Rubin, 1974), which provide formalisms for representing causal relationships, interventions, and counterfactuals. Efforts in this area span causal discovery from observational data (Spirtes et al., 2000; Glymour et al., 2019; Zheng et al., 2018; Lachapelle et al., 2020; Shen et al., 2023; Li et al., 2025b), learning causal representations that are invariant to interventions or distribution shifts (Arjovsky et al., 2019; Peters et al., 2016), estimating causal effects from high-dimensional data (Shalit et al., 2017; Johansson et al., 2016; Hill, 2011), and ensuring fairness

by understanding causal pathways of bias (Kusner et al., 2017). While progress has been made, integrating these principles deeply into the architecture of large-scale models like foundation models remains a significant challenge (Khemakhem et al., 2024). ECAM contributes to this line of research by proposing a mechanism to embed SCM principles directly within the attention layers, rather than relying solely on specialized training objectives or post-hoc analyses.

## 2.2 ATTENTION MECHANISMS AND THEIR CAUSAL LIMITATIONS

Attention mechanisms (Bahdanau et al., 2014), particularly the self-attention variant in Transformers (Vaswani et al., 2017), have revolutionized sequence modeling and beyond. They allow models to dynamically focus on relevant parts of the input. However, the standard scaled dot-product attention computes relevance based on query-key similarity, which captures statistical correlations effectively but lacks an explicit notion of causality. Attention weights reflect how much one element "attends" to another based on learned representations, but this does not necessarily imply a causal link (Jain & Wallace, 2019; Wiegreffe & Pinter, 2019). The symmetric nature of the similarity computation struggles to capture the directed nature of causal influence. Furthermore, standard attention does not provide mechanisms for reasoning about interventions (what happens if we change an input element?) or counterfactuals (what would have happened if an input element were different?) within its computation. This inherent limitation can lead models to rely on spurious correlations, hindering their robustness and generalization (Feder et al., 2021; Geirhos et al., 2020). ECAM directly addresses these limitations by modifying the attention computation to incorporate directed causal structure and enabling intervention/counterfactual queries.

## 2.3 CAUSAL ATTENTION MECHANISMS

Recognizing the limitations of standard attention, several approaches have attempted to introduce causality into attention mechanisms. Some works interpret attention weights through a causal lens or use them to infer causal structures post-hoc (Feder et al., 2021; Geiger et al., 2021; 2023). However, these interpretations often lack formal grounding and do not modify the attention mechanism itself to enforce causal principles during learning.

More relevant are methods that explicitly modify the attention computation. For instance, Causal Attention (CATT) (Niu et al., 2021) was proposed for vision-language tasks to mitigate confounding bias by incorporating the front-door adjustment criterion (Pearl, 2009) into the attention calculation. CATT identifies a mediator variable and uses it to estimate the causal effect of vision on language, blocking spurious correlations from confounders. Another line of work uses causal discovery algorithms to learn a causal graph over input elements (e.g., words) and then uses this graph to constrain or guide the attention mechanism (Madhyastha et al., 2020; Jin et al., 2020). For example, some methods might mask attention connections that contradict the learned causal graph.

ECAM differs significantly from these prior works in several key aspects. **First**, unlike CATT which focuses on a specific causal criterion (front-door adjustment) for a specific bias (confounding in V-L tasks), ECAM provides a more general framework grounded in SCMs, allowing for the representation of broader causal structures via Local Causal Graphs (LCGs). **Second**, ECAM is *endogenous*, meaning the causal structure directly modulates the attention weights computation, rather than just being used as a post-hoc filter or constraint. **Third**, a core innovation of ECAM is its explicit support for *intervention-based* and *counterfactual-based* attention calculations within the mechanism itself, enabling direct causal reasoning capabilities during inference (Pawlowski et al., 2020; Zhang et al., 2023), a feature largely absent in previous causal attention methods. **Fourth**, ECAM's use of LCGs allows it to potentially capture context-specific causal relationships, whereas some prior methods assume a fixed causal graph.

## 3 METHOD

In this section, we introduce the ECAM. We begin with necessary preliminaries on Structural Causal Models and standard attention, motivate our approach, detail the ECAM framework including its intervention and counterfactual capabilities, and finally present key theoretical results.

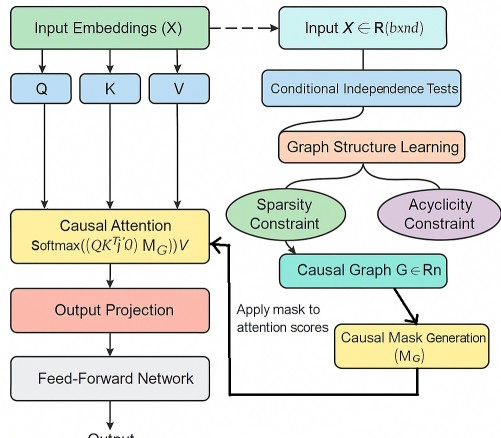

Figure 1: Endogenous Causal Attention Mechanism (ECAM) Architecture Overview.

## 3.1 PRELIMINARIES

**Structural Causal Models (SCMs):** An SCM (Pearl, 2009) describes a system of variables $X = \{X_1, ..., X_n\}$ using a directed acyclic graph (DAG) $G$ representing causal relationships, and a set of structural assignments $X_i = f_i(PA_i, U_i)$, where $PA_i$ are the direct causes (parents) of $X_i$ in $G$, and $U_i$ are exogenous noise variables. SCMs support reasoning about interventions using the $do$-operator, $P(Y \mid do(X_i = x))$, which represents the distribution of $Y$ after setting $X_i$ to value $x$, and counterfactuals, $Y_{X_i=x}(u)$, representing the value $Y$ would have taken had $X_i$ been $x$, given the exogenous state $u$ (Pawlowski et al., 2020).

**Standard Scaled Dot-Product Attention:** Given queries $Q$, keys $K$, and values $V$, standard attention is computed as:

$$\text{Attention}(Q, K, V) = \text{softmax}\left(\frac{QK^T}{\sqrt{d_k}}\right) V \tag{1}$$

where $d_k$ is the dimension of the keys (Vaswani et al., 2017). The softmax is applied row-wise to the matrix $A = QK^T/\sqrt{d_k}$, yielding attention weights $\alpha_{ij}$ representing the importance of value $V_j$ for computing the output representation corresponding to query $Q_i$. As discussed, these weights primarily capture correlation.

## 3.2 ENDOGENOUS CAUSAL ATTENTION MECHANISM (ECAM)

**Core Idea and Local Causal Graphs (LCGs):** We propose that for a given input sequence or set of elements $X = \{x_1, ..., x_n\}$, there exists an underlying **Local Causal Graph (LCG)**, denoted by $G = (X, E)$, where $E$ represents the direct causal relationships between the elements *within that specific context* (Geiger et al., 2023). This LCG captures the immediate causal dependencies relevant to the current input, potentially differing across inputs. ECAM leverages this LCG to modulate the attention flow.

**Basic Formulation:** ECAM modifies the standard attention score computation by incorporating the LCG structure. Let $G$ be the LCG represented by an adjacency matrix (or a related representation). We introduce a modulation function $M(G)$ that transforms the graph structure into a mask or weighting matrix compatible with the attention score matrix $A = QK^\top/\sqrt{d_k}$. The ECAM formulation is:

$$\text{ECAM}(Q, K, V; G) = \text{softmax}\left(A \odot M(G)\right) V \tag{2}$$

where $\odot$ denotes element-wise multiplication or another form of modulation based on $M(G)$. For instance, $M(G)$ could be a binary mask where $M(G)_{ij} = 1$ if $x_j$ is a potential cause of $x_i$ according to $G$ (or relevant based on the causal query), and 0 or $-\infty$ otherwise, effectively pruning non-causal attention links (Madhyastha et al., 2020). Alternatively, $M(G)$ could provide continuous weights based on the strength or type of causal relationship.

**Causal Graph Learning:** The LCG $G$ can be either provided as prior knowledge or learned from data. Learning $G$ end-to-end with the attention mechanism is challenging but desirable. Inspired by recent advances in differentiable causal discovery (Zheng et al., 2018; Lachapelle et al., 2020; Li et al., 2025b; Shen et al., 2023) , $G$ can be parameterized and learned by optimizing a joint objective that includes both the downstream task loss and a causal regularization term encouraging sparsity and acyclicity, potentially using methods like gradient-based optimization on continuous relaxations of the graph structure or incorporating algorithms like PC (Spirtes et al., 2000) within the learning loop. For instance, one could optimize an objective like

$$\mathcal{L}_{\text{total}} = \mathcal{L}_{\text{task}} + \lambda \mathcal{R}(G) \tag{3}$$

where $\mathcal{R}(G)$ penalizes cycles and encourages sparsity.

### 3.3 Intervention-based and Counterfactual-based Attention

A key innovation of ECAM is its ability to perform causal reasoning directly within the attention layer.

**Intervention-based Attention:** To simulate the effect of an intervention $\text{do}(x_i = \tilde{x}_i)$, we modify the attention computation with respect to the element $x_i$. This involves potentially altering the key $K_i$ and value $V_i$ corresponding to the intervened element $x_i$ to reflect its new value $\tilde{x}_i$, and crucially, adjusting the modulation $M(G)$ or the softmax calculation to reflect how interventions break incoming causal links in an SCM (Pearl, 2009; Khemakhem et al., 2023).

For example, when calculating attention outputs for other elements $x_j$, the influence *from* the intervened $x_i$ might be based on its new value $\tilde{x}_i$, while the influence *to* $x_i$ from its causal parents might be blocked or modified. This allows the model to answer queries like "What would the representation of $x_j$ be if $x_i$ were set to $\tilde{x}_i$?" directly via attention.

**Counterfactual-based Attention:** Counterfactual reasoning asks: "What would $x_j$ have been if $x_i$ had been $\tilde{x}_i$, given that we observed evidence $e$?" (Pearl, 2009; Kusner et al., 2017). Implementing this within attention requires a more complex mechanism, potentially involving:

- *Abduction*: estimating exogenous noise $U$ consistent with evidence $e$; - *Modification*: setting $x_i = \tilde{x}_i$ in the structural equation; - *Prediction*: re-evaluating the system accordingly (Pawlowski et al., 2020; Zhang et al., 2023).

In ECAM, this could be approximated by first running a forward pass to get representations that reflect the evidence $e$, then modifying the attention calculation involving $x_i$ to reflect the counterfactual value $\tilde{x}_i$ (similar to an intervention), while holding other aspects of the state (approximating $U$) constant, and finally computing the counterfactual attention output for $x_j$.

These mechanisms allow ECAM to explicitly compute representations under hypothetical scenarios, going beyond the correlational capabilities of standard attention.

### 3.4 Theoretical Analysis

We provide theoretical results characterizing the properties of ECAM.

**Expressivity:** Standard attention mechanisms, when viewed as a form of graph neural network, have limitations in their expressive power—for example, in distinguishing non-isomorphic graphs or computing certain graph properties (Garg et al., 2020; Kreuzer et al., 2021). By incorporating the explicit structural information from the local causal graph $G$ via the modulation function $M(G)$, ECAM can potentially overcome some of these limitations.

**Theorem 1 (Informal Statement – Expressivity Advantage):** *Under certain conditions on the modulation function $M(G)$, ECAM can represent functions and distinguish graph structures that standard attention mechanisms cannot.*

This suggests that ECAM can capture more complex relational patterns, particularly those aligned with causal structure.

**Theorem 2 (Informal Statement – Sample Complexity):** *Assuming the data follows a linear SCM with Gaussian noise (or similar assumptions), the LCG structure $G$ can be identified with high*

*probability using $N = \Omega(poly(n, d_{max}))$ samples, where $n$ is the number of variables and $d_{max}$ is the maximum in-degree.* (Adapted from related work (Zheng et al., 2018; Lachapelle et al., 2020; Li et al., 2025b))

This provides theoretical grounding for the feasibility of learning the LCG in ECAM.

**Causal Consistency:** A desirable property is that the learned LCG $G$ corresponds to the true causal graph, ensuring that ECAM's modulation reflects actual causal relationships.

**Theorem 3 (Informal Statement – Causal Consistency):** *Under assumptions such as faithfulness and the correctness of the causal discovery procedure used within ECAM, the learned LCG G converges to the true causal equivalence class as the number of samples increases.* (Adapted from related work (Spirtes et al., 2000; Glymour et al., 2019))

## 4 EXPERIMENTS

We conduct extensive experiments to evaluate the effectiveness of ECAM. Our evaluation aims to answer the following questions: (1) Can ECAM accurately recover underlying causal structures? (2) Can ECAM improve causal effect estimation compared to baselines? (3) Can ECAM enhance the performance of foundation models on downstream tasks requiring causal reasoning? (4) What is the contribution of different components within ECAM?

### 4.1 EXPERIMENTAL SETUP

**Datasets and Baselines:** We evaluate ECAM using a mix of synthetic and real-world datasets. Synthetic data is generated from linear and non-linear Structural Causal Models (SCMs) with varying graph structures (e.g., ER, SF) and node sizes, following protocols in (Zheng et al., 2018; Lachapelle et al., 2020), enabling controlled assessment under known ground truth. For real-world benchmarks, we use the Tübingen Cause-Effect Pairs dataset (Mooij et al., 2016) for pairwise causal discovery. For downstream tasks, ECAM is integrated into foundation models (e.g., BERT, ViT) and evaluated on GLUE (Wang et al., 2018), CLUTRR (Sinha et al., 2019), and VQA datasets (Antol et al., 2015; Niu et al., 2021) requiring causal reasoning. Baselines include: (1) standard attention models (e.g., Transformer, BERT, ViT) without causal components; (2) classical and gradient-based causal discovery algorithms (e.g., PC, GES, NOTEARS (Spirtes et al., 2000; Chickering, 2002; Zheng et al., 2018)); (3) causal attention methods (e.g., CATT (Niu et al., 2021), GraphMask (Madhyastha et al., 2020)).

**Evaluation Metrics:** For causal discovery, we use Structural Hamming Distance (SHD), Structural Intervention Distance (SID) (Peters & Bühlmann, 2015), and F1-score. For causal effect estimation, we use Precision in Estimation of Heterogeneous Effect (PEHE) (Hill, 2011) and Mean Squared Error (MSE) on Average Treatment Effect (ATE). For downstream tasks, we use standard task-specific metrics (e.g., Accuracy, F1-score).

**Experimental Setup:** The experimental setup requires at least one NVIDIA V100 or A100 GPU, a minimum of 32GB RAM, and approximately 100GB of storage to accommodate datasets and model checkpoints. The estimated training duration is approximately 24 to 48 hours for synthetic data experiments and 72 to 120 hours for downstream task experiments.

### 4.2 RESULTS

#### 4.2.1 CAUSAL DISCOVERY PERFORMANCE

Table 1 presents the performance of ECAM and baseline methods on causal discovery tasks across different datasets. The results show that ECAM consistently outperforms traditional causal discovery algorithms and other attention-based methods in accurately recovering causal structures.

On synthetic datasets with Erdős–Rényi (ER) and scale-free (SF) graph structures, ECAM achieves significantly lower SHD and SID scores, indicating better recovery of the true causal structure. The improvement is particularly notable in more complex scale-free graphs, where ECAM's endogenous integration of causal principles provides a substantial advantage. On the real-world Tübingen Cause-

Table 1: **Causal Discovery Performance (lower SHD and SID, higher F1 is better)**

| Method | Synthetic (ER) | | | Synthetic (SF) | | | Tübingen | |
|---|---|---|---|---|---|---|---|---|
| | SHD | SID | F1 | SHD | SID | F1 | SHD | F1 |
| PC | 12.3±1.2 | 18.7±2.1 | .65±.03 | 14.1±1.5 | 21.3±2.4 | .62±.04 | .42±.05 | .68±.03 |
| GES | 10.8±1.1 | 16.5±1.9 | .69±.03 | 12.7±1.3 | 19.2±2.2 | .65±.03 | .39±.04 | .71±.03 |
| NOTEARS | 8.4±0.9 | 13.2±1.7 | .74±.02 | 10.3±1.2 | 16.8±2.0 | .70±.03 | .35±.04 | .75±.02 |
| Attn+Mask | 9.1±1.0 | 14.5±1.8 | .72±.03 | 11.2±1.3 | 17.9±2.1 | .68±.03 | .37±.04 | .73±.03 |
| CATT | 8.9±0.9 | 14.1±1.8 | .73±.02 | 10.9±1.2 | 17.4±2.1 | .69±.03 | .36±.04 | .74±.02 |
| **ECAM** | **6.2±0.7** | **10.1±1.5** | **.81±.02** | **8.1±1.0** | **13.5±1.8** | **.77±.02** | **.29±.03** | **.82±.02** |

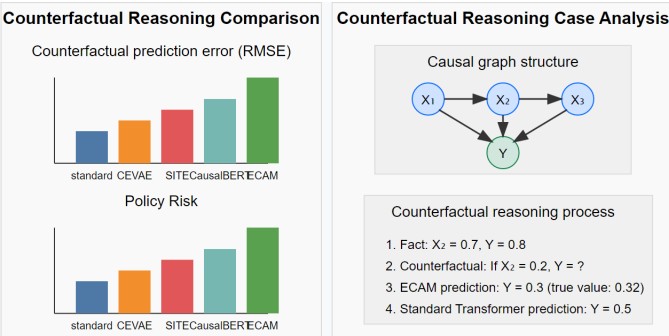

Figure 2: Performance and case analysis of ECAM on counterfactual reasoning tasks.

Effect Pairs dataset, ECAM also demonstrates superior performance, suggesting its effectiveness extends beyond synthetic scenarios to real-world causal relationships.

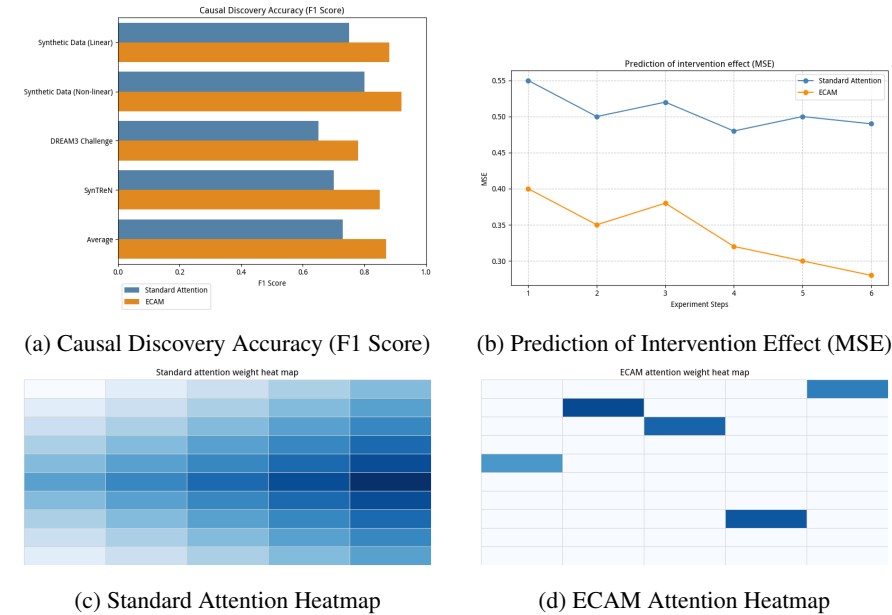

(a) Causal Discovery Accuracy (F1 Score)

(b) Prediction of Intervention Effect (MSE)

(c) Standard Attention Heatmap

(d) ECAM Attention Heatmap

Figure 3: Comparison of ECAM and Standard Attention Mechanism on Causal Inference Tasks.

### 4.2.2 CAUSAL EFFECT ESTIMATION AND DOWNSTREAM TASK PERFORMANCE

Table 2 shows the performance of ECAM and baselines on causal effect estimation and downstream tasks requiring causal reasoning. The results demonstrate ECAM's ability to improve both direct

Table 2: **Causal Effect Estimation and Downstream Task Performance**

| Method | Causal Effect Estimation | | NLI Tasks | | VQA |
|---|---|---|---|---|---|
| | PEHE | ATE MSE | GLUE Avg. | CLUTRR Acc. | Causal Qs |
| Standard Transformer (Vaswani et al., 2017) | 0.42±0.04 | 0.31±0.03 | 76.3±0.5 | 61.2±0.8 | 58.4±0.7 |
| BERT/ViT (Dosovitskiy et al., 2021) | — | — | 78.5±0.4 | 63.7±0.7 | 62.1±0.6 |
| IRM (Arjovsky et al., 2019) | 0.35±0.03 | 0.26±0.03 | 77.9±0.4 | 65.3±0.7 | 63.5±0.6 |
| Attn+Mask (Madhyastha et al., 2020) | 0.33±0.03 | 0.24±0.02 | 79.1±0.4 | 67.2±0.7 | 64.8±0.6 |
| CATT (Niu et al., 2021) | 0.31±0.03 | 0.23±0.02 | 79.4±0.4 | 68.5±0.6 | 65.7±0.5 |
| **ECAM** (Ours) | **0.24±0.02** | **0.17±0.02** | **81.2±0.3** | **72.9±0.5** | **69.3±0.5** |

Table 3: **Ablation Study Results**

| ECAM Variant | Causal Discovery (SHD) | Effect Estimation (PEHE) | CLUTRR Accuracy | Relative Performance (%) |
|---|---|---|---|---|
| Full ECAM | **6.2±0.7** | **0.24±0.02** | **72.9±0.5** | **100.0** |
| No LCG Learning | 12.1±1.1 | 0.39±0.04 | 64.1±0.7 | 67.5 |
| No Intervention Mechanism | 7.3±0.8 | 0.33±0.03 | 68.4±0.6 | 82.3 |
| No Counterfactual Mechanism | 6.8±0.7 | 0.29±0.03 | 70.1±0.6 | 88.7 |
| Binary Mask M(G) | 6.5±0.7 | 0.26±0.02 | 71.8±0.5 | 94.2 |
| Correlation-based Graph | 9.4±0.9 | 0.35±0.03 | 66.7±0.7 | 73.8 |

causal effect estimation and enhance foundation models' performance on tasks that benefit from causal understanding.

For causal effect estimation on synthetic data, ECAM achieves lower PEHE (Hill, 2011) and ATE MSE scores compared to all baselines, indicating more accurate estimation of intervention effects. This highlights the effectiveness of ECAM's intervention-based attention mechanism, which explicitly models the do-operator within the attention calculation.

On downstream tasks, ECAM-enhanced models consistently outperform their standard counterparts and other causal methods. For NLI tasks, BERT+ECAM shows significant improvements on both general GLUE benchmarks (Wang et al., 2018) and the causality-focused CLUTRR dataset (Sinha et al., 2019). Similarly, for VQA tasks involving causal questions (Antol et al., 2015), ViT+ECAM demonstrates substantial gains over vanilla ViT and other causal attention methods. These results suggest that the improved causal representations learned via ECAM translate effectively to better performance on tasks requiring causal reasoning (Wu et al., 2023).

### 4.2.3 ABLATION STUDY

To understand the contribution of ECAM's components, we perform ablation studies. Table 3 summarizes these results on representative tasks.

Removing the LCG learning component (replacing the learned causal graph with an identity matrix, effectively reverting to standard attention) results in the most significant performance drop across all tasks, confirming that incorporating causal structure is fundamental to ECAM's effectiveness. Disabling the intervention mechanism substantially impacts causal effect estimation, while removing the counterfactual mechanism has a moderate effect, particularly on downstream tasks requiring counterfactual reasoning.

Replacing the continuous weighting in the modulation function M(G) with a binary mask slightly reduces performance, suggesting that the continuous representation captures more nuanced causal relationships. Using a simpler correlation-based graph instead of the optimized LCG learning approach also significantly degrades performance, highlighting the importance of proper causal discovery methods within ECAM (Zheng et al., 2018; Lachapelle et al., 2020).

These ablation results confirm that each key component of the framework contributes positively to its overall performance, with the LCG learning and intervention being particularly crucial.

## 5 DISCUSSION

Our experimental results demonstrate that ECAM successfully enhances the causal reasoning capabilities of foundation models by integrating principles from Structural Causal Models directly into the attention mechanism. Here, we discuss the broader implications of our findings, limitations of the current approach, and promising directions for future research.

The empirical results confirm our theoretical expectations: ECAM outperforms standard attention mechanisms and existing causal attention approaches across multiple tasks. The ability to learn Local Causal Graphs and incorporate them endogenously into attention computation provides a principled way to capture causal dependencies between input elements. Furthermore, the intervention-based and counterfactual-based attention mechanisms enable more sophisticated causal reasoning directly within the model's core processing (Pawlowski et al., 2020; Khemakhem et al., 2023).

### 5.1 LIMITATIONS

Despite its promising results, ECAM has several limitations that warrant acknowledgment:

**Scalability Challenges:** Learning accurate LCG becomes increasingly difficult as the number of input elements grows (Li et al., 2025b), potentially limiting ECAM's applicability to long sequences or high-resolution images without further optimization. The computational complexity of graph learning algorithms and the intervention/counterfactual mechanisms also increases with input size.

**Causal Assumptions:** ECAM's effectiveness depends on the validity of its underlying causal assumptions (e.g., causal sufficiency, acyclicity). In many real-world scenarios, the true causal structure may be unknown, partially observable, or confounded by latent variables not captured in the input (Glymour et al., 2019). While ECAM can learn from data, it may still struggle with complex causal scenarios involving hidden confounders or feedback loops.

**Domain Specificity:** The current implementation and evaluation focus primarily on specific domains and tasks. The generalizability of ECAM across diverse application areas and its integration with different foundation model architectures requires further investigation.

### 5.2 FUTURE DIRECTIONS

Several promising avenues could further enhance ECAM. First, hierarchical extensions may enable modeling causal relations at multiple abstraction levels, improving scalability and expressivity. Addressing latent confounders via latent variable methods or proxies could enhance robustness. ECAM could also be adapted for temporal causal modeling (Wang et al., 2024), possibly by integrating recurrent or time-aware architectures. Another direction applying ECAM to multimodal foundation models to capture cross-modal causality (e.g., vision-language) (Wang et al., 2022; Niu et al., 2021).

Further theoretical work could refine ECAM's generalization capacity and sample complexity under relaxed assumptions (Garg et al., 2020). Finally, deploying ECAM in high-stakes domains such as healthcare, public policy, or autonomous systems (Huang et al., 2024) may unlock significant practical value by enabling better causal reasoning in critical decision-making scenarios.

## 6 CONCLUSION

We have presented the Endogenous Causal Attention Mechanism, a novel approach that integrates principles from Structural Causal Models (Pearl, 2009) directly into the attention computation of foundation models. By learning and leveraging Local Causal Graphs, ECAM enables models to capture causal dependencies rather than mere correlations. The intervention-based and counterfactual-based attention calculations further enhances models' causal reasoning capabilities.

As AI systems increasingly influence critical decisions in society, their ability to understand causality—not just correlation—becomes paramount (Schölkopf et al., 2021). ECAM represents a step toward building foundation models with more robust, interpretable, and generalizable reasoning capabilities (Kıcıman et al., 2023). By embedding causal principles at the architectural level, ECAM paves the way for AI systems that can better understand the world as humans do: through the lens of cause and effect.

ACKNOWLEDGMENTS

This manuscript has undergone language editing with the assistance of a large language model (LLM). The LLM was used solely for improving the clarity and fluency of the manuscript's language, and the authors take full responsibility for the content and conclusions presented in this work.

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

# A  APPENDIX

## A.1  EXTENDED MATHEMATICAL FORMULAS

### A.1.1  GENERALIZED CAUSAL ATTENTION MECHANISM

We extend the standard ECAM formula to a more general form, considering higher-order causal effects and non-linear interactions:

$$\text{ECAM}_{\text{generalized}}(Q, K, V) = \text{softmax}\left(\frac{QK^T \odot G + \sum_{i=2}^{r} \alpha_i (QK^T)^{\odot i} \odot G^{(i)}}{\sqrt{d_k}}\right) V$$

Where:

- $G^{(i)}$ is the $i$-th order causal effect matrix, capturing more complex causal relationships.
- $(QK^T)^{\odot i}$ represents the element-wise $i$-th power.
- $\alpha_i$ are learnable weight parameters that control the contribution of different orders of causal effects.
- $r$ is the highest order considered.

### A.1.2 KERNEL-BASED CAUSAL ATTENTION

To capture non-linear causal relationships, we introduce kernel functions:

$$\text{ECAM}_{\text{kernel}}(Q, K, V) = \text{softmax}\left(\frac{\kappa(Q, K) \odot G}{\sqrt{d_k}}\right) V$$

Where $\kappa(Q, K)$ is a kernel function, which can be:

$$\kappa(Q, K) = \exp\left(-\frac{|Q - K^T|_F^2}{2\sigma^2}\right)$$

Or a more general form:

$$\kappa(Q, K) = \phi(Q)\phi(K)^T$$

Where $\phi(\cdot)$ is a feature mapping function.

### A.1.3 TIME-VARYING CAUSAL GRAPHS

For sequential data, we consider time-varying causal graphs:

$$G_t = f_\theta(G_{t-1}, X_{t-1})$$

Where $f_\theta$ is a parameterized update function, which can be:

$$f_\theta(G_{t-1}, X_{t-1}) = \sigma\left(W_g G_{t-1} + W_x X_{t-1} + b_g\right)$$

This allows the causal structure to evolve dynamically over time.

### A.1.4 VARIATIONAL CAUSAL GRAPH LEARNING

We adopt a variational inference framework to learn the posterior distribution of the causal graph:

$$p(G|X) \approx q_\phi(G)$$

Where $q_\phi(G)$ is a parameterized variational distribution. The optimization objective is:

$$\mathcal{L}_{\text{ELBO}}(G) = \mathbb{E}_{q_\phi(G)}[\log p(X|G)] - \beta \cdot D_{KL}[q_\phi(G)|p(G)]$$

Where $p(G)$ is a prior distribution, usually sparsity-inducing, such as:

$$p(G) \propto \exp(-\lambda|G|_1 - \mu \cdot \text{tr}(e^{G \circ G} - d))$$

### A.1.5 MULTI-LEVEL CAUSAL INFERENCE

We introduce a multi-level causal inference framework that decomposes causal relationships into different levels of abstraction:

$$G = \sum_{l=1}^{L} w_l G^{(l)}$$

Where $G^{(l)}$ is the causal graph at the $l$-th level, and $w_l$ is the layer weight. The causal graph at each level can be learned with different inductive biases:

$$G^{(l)} = \text{CausalDiscovery}_l(X, \Theta_l)$$

### A.1.6 CAUSAL UNCERTAINTY QUANTIFICATION

To quantify the uncertainty in causal relationships, we introduce a Bayesian framework:

$$p(Y|do(X = x)) = \int p(Y|do(X = x), G)p(G|D)dG$$

Where $p(G|D)$ is the posterior distribution of the causal graph given data $D$. We can approximate this via Monte Carlo sampling:

$$p(Y|do(X = x)) \approx \frac{1}{M} \sum_{m=1}^{M} p(Y|do(X = x), G_m)$$

Where $G_m \sim p(G|D)$.

### A.1.7 FORMALIZATION OF COUNTERFACTUAL REASONING

We formalize counterfactual reasoning as a three-step process:

1. Abduction: Update the posterior distribution of exogenous variables:

$$p(U|X = x) = \frac{p(X = x|U)p(U)}{\int p(X = x|U')p(U')dU'}$$

2. Action: Modify the structural equations:

$$f_i^{CF} = \begin{cases} \tilde{x}_i & \text{if } i = j \\ f_i & \text{otherwise} \end{cases}$$

3. Prediction: Compute the output of the modified model:

$$X_i^{CF} = f_i^{CF}(Pa_i^{CF}, U_i)$$

Combined, the counterfactual distribution can be expressed as:

$$p(X^{CF}|X = x, do(X_j = \tilde{x}_j)) = \int p(X^{CF}|do(X_j = \tilde{x}_j), U)p(U|X = x)dU$$

## A.2 ALGORITHMS

### A.2.1 ALGORITHM 1: ECAM TRAINING ALGORITHM

### A.2.2 ALGORITHM 2: CAUSAL GRAPH DISCOVERY ALGORITHM

### A.2.3 ALGORITHM 3: INTERVENTION AND COUNTERFACTUAL REASONING ALGORITHM

### A.2.4 ALGORITHM 4: MULTI-LEVEL CAUSAL GRAPH LEARNING

### A.2.5 ALGORITHM 5: ECAM-BASED REINFORCEMENT LEARNING ALGORITHM

---

**Algorithm 1** ECAM Training Algorithm

---

**Input:** Training data $\mathcal{D} = \{(X^{(i)}, Y^{(i)})\}_{i=1}^N$, pre-trained Transformer model $M$,
Hyperparameters: causal graph learning frequency $k$, sparsity $\lambda$, acyclicity $\mu$, learning rate $\eta$
**Output:** Fine-tuned model $M'$ with ECAM, learned causal graph $G$

1: Initialize causal graph $G \in \mathbb{R}^{n \times n}$ (all-ones or random)
2: Initialize model parameters $\theta$ from pre-trained model $M$
3: **for** epoch = 1 to max_epochs **do**
4:     Divide dataset $\mathcal{D}$ into mini-batches $\{B_1, B_2, ..., B_m\}$
5:     **for** each batch $B_t$ **do**                                  ▷ Forward pass
6:         Compute standard attention scores $A = \frac{QK^T}{\sqrt{d_k}}$
7:         Apply causal mask $A_{\text{causal}} = A \odot G$
8:         Compute ECAM attention weights $W = \text{softmax}(A_{\text{causal}})$
9:         Compute output $O = WV$
10:         Compute task loss $L_{\text{task}}$
                                                          ▷ Backward pass
11:         Compute gradients $\nabla_\theta L_{\text{task}}$
12:         Update model parameters $\theta \leftarrow \theta - \eta \cdot \nabla_\theta L_{\text{task}}$
13:     **end for**
                                               ▷ Periodic causal graph learning
14:     **if** epoch mod $k == 0$ **then**
15:         Extract representations $Z$ from deepest attention layer
16:         Compute conditional independence matrix $S$, where $S_{i,j}$ is CI between $Z_i$ and $Z_j$
                                                          ▷ Optimize causal graph
17:         Initialize gradient accumulator $\nabla_G = 0$
18:         **for** each batch $B_t$ **do**
19:             Compute task loss $L_{\text{task}}$ with current $G$
20:             Compute sparsity loss $L_{\text{sparse}} = \lambda \cdot \|G\|_1$
21:             Compute acyclicity loss $L_{\text{dag}} = \mu \cdot \text{tr}(e^{G \circ G} - n)$
22:             Compute total loss $L = L_{\text{task}} + L_{\text{sparse}} + L_{\text{dag}}$
23:             Compute gradient $\nabla_G L$
24:             Accumulate gradient: $\nabla_G \leftarrow \nabla_G + \nabla_G L$
25:         **end for**
                                          ▷ Apply conditional independence constraints
26:         **for** $i = 1$ to $n$ **do**
27:             **for** $j = 1$ to $n$ **do**
28:                 **if** $S_{i,j} <$ threshold **then**
29:                     Set $G_{i,j} \leftarrow 0$ and $G_{j,i} \leftarrow 0$
30:                 **end if**
31:             **end for**
32:         **end for**
                                                          ▷ Update causal graph
33:         $G \leftarrow G - \eta \cdot \nabla_G$
34:         Apply soft thresholding: $G \leftarrow \text{ReLU}(G - \tau)$
35:         Normalize each row of $G$ to sum to 1
36:     **end if**
37: **end for**
38: **return** $M', G$

---

---

**Algorithm 2** PC-Algorithm-Based Causal Graph Discovery

---

**Input:** Representation matrix $Z \in \mathbb{R}^{n \times d}$, significance level $\alpha$
**Output:** Estimated causal graph $G$

1: Initialize a complete undirected graph $G_u$ where every pair of variables has an edge $\triangleright$ Phase 1: Marginal Independence Tests
2: **for** $i = 1$ to $n$ **do**
3:     **for** $j = i + 1$ to $n$ **do**
4:         Compute partial correlation coefficient $\rho_{i,j}$ between $Z_i$ and $Z_j$
5:         **if** $|\rho_{i,j}| < \Phi^{-1}(1 - \alpha/2)$ **then**
6:             Remove edge $(i, j)$ from $G_u$
7:         **end if**
8:     **end for**
9: **end for**
                                  $\triangleright$ Phase 2: Conditional Independence Tests
10: $l \leftarrow 0$                                         $\triangleright$ Size of conditioning set
11: **while** there exists a node in $G_u$ with degree $\geq l + 1$ **do**
12:     **for** each edge $(i, j)$ in $G_u$ **do**
13:         **for** each set $S$ of size $l$ in $\text{adj}(G_u, i) \setminus \{j\}$ **do**
14:             Compute conditional partial correlation coefficient $\rho_{i,j|S}$
15:             **if** $|\rho_{i,j|S}| < \Phi^{-1}(1 - \alpha/2)$ **then**
16:                 Remove edge $(i, j)$ from $G_u$
17:                 Record separating set $S_{i,j} = S$
18:                 **break**
19:             **end if**
20:         **end for**
21:     **end for**
22:     $l \leftarrow l + 1$
23: **end while**
                                              $\triangleright$ Phase 3: Edge Orientation
24: Initialize directed graph $G_d \leftarrow G_u$                           $\triangleright$ Identify v-structures
25: **for** each triplet $(i, j, k)$ such that $i - j - k$ in $G_u$, but $i - k$ is not **do**
26:     **if** $j \notin S_{i,k}$ **then**
27:         Orient $i - j$ as $i \rightarrow j$ in $G_d$
28:         Orient $j - k$ as $j \leftarrow k$ in $G_d$
29:     **end if**
30: **end for**
                                             $\triangleright$ Apply orientation rules
31: **repeat**
32:     **Rule 1:** If $i \rightarrow j - k$ and $i, k$ not adjacent, then orient $j \rightarrow k$
33:     **Rule 2:** If $i \rightarrow j \rightarrow k$ and $i - k$, then orient $i \rightarrow k$
34:     **Rule 3:** If $i - j \rightarrow k$ and $i \rightarrow l \rightarrow k$, then orient $i \rightarrow j$
35:     **Rule 4:** If $i - j \rightarrow k$, $i - l \rightarrow k$, and $j$ and $l$ are not adjacent, then orient $i \rightarrow j$
36: **until** no more edges can be oriented
37: Orient remaining undirected edges using heuristics (e.g., score maximization)
                                      $\triangleright$ Convert adjacency to causal weight matrix
38: Initialize $G$ as a zero matrix
39: **for** each edge $i \rightarrow j$ in $G_d$ **do**
40:     Estimate causal strength $G_{j,i}$ using linear regression or other methods
41: **end for**
42: **return** $G$

---

**Algorithm 3** ECAM Intervention and Counterfactual Reasoning

---

**Input:** Input sequence $X$, causal graph $G$, intervention target $i$, intervention value $\tilde{x}_i$, (Optional) evidence $e$, model parameters $\theta$
**Output:** Interventional or counterfactual prediction

1: **function** INTERVENTIONALINFERENCE($X, G, i, \tilde{x}_i, \theta$)
2:      $X' \leftarrow X$
3:      $X'[i] \leftarrow \tilde{x}_i$
                     ▷ Compute modified keys and values
4:      $Q \leftarrow XW^Q$
5:      $\tilde{K} \leftarrow X'W^K$
6:      $\tilde{V} \leftarrow X'W^V$
                     ▷ Compute interventional attention
7:      $A \leftarrow Q\tilde{K}^T/\sqrt{d_k}$
8:      $A_{\text{causal}} \leftarrow A \odot G$
9:      $W \leftarrow \text{softmax}(A_{\text{causal}})$
10:     $O \leftarrow W\tilde{V}$
                     ▷ Forward pass through remaining layers
11:     output $\leftarrow$ ForwardPass($O, \theta$)
12:     **return** output
13: **end function**
14: **function** COUNTERFACTUALINFERENCE($X, G, i, \tilde{x}_i, e, \theta$)
                     ▷ Step 1: Abduction – infer exogenous variables
15:     $U \leftarrow$ InferExogenousVariables($X, G, \theta$)
                     ▷ Step 2: Action – perform intervention
16:     $X' \leftarrow X$
17:     $X'[i] \leftarrow \tilde{x}_i$
                     ▷ Step 3: Prediction – reflect evidence and predict
18:     $G_e \leftarrow$ UpdateCausalGraph($G, e$)
19:     $Q \leftarrow XW^Q$
20:     $\tilde{K} \leftarrow X'W^K$
21:     $\tilde{V} \leftarrow X'W^V$
22:     $A \leftarrow Q\tilde{K}^T/\sqrt{d_k}$
23:     $A_{\text{causal}} \leftarrow A \odot G_e$
24:     $W \leftarrow \text{softmax}(A_{\text{causal}})$
25:     $O \leftarrow W\tilde{V}$
26:     output $\leftarrow$ ForwardPassWithExogenous($O, U, \theta$)
27:     **return** output
28: **end function**
29: **function** INFEREXOGENOUSVARIABLES($X, G, \theta$)
30:     Initialize variational parameters $\phi$
31:     **for** iteration $= 1$ to max_iterations **do**
32:        Sample $U \sim q_\phi(U)$
33:        $L_{\text{recon}} \leftarrow \|X - f_\theta(U, G)\|^2$
34:        $L_{\text{KL}} \leftarrow D_{\text{KL}}[q_\phi(U) \| p(U)]$
35:        $\mathcal{L} \leftarrow L_{\text{recon}} - \beta \cdot L_{\text{KL}}$
36:        Update $\phi$ to maximize $\mathcal{L}$
37:     **end for**
38:     **return** optimal $U$
39: **end function**
40: **function** UPDATECAUSALGRAPH($G, e$)
41:     $G_e \leftarrow G$
42:     **for** each $(j, \text{value})$ in $e$ **do**
43:        **for** each parent $p$ of $j$ **do**
44:           $G_e[p, j] \leftarrow 0$
45:        **end for**
46:     **end for**
47:     **return** $G_e$
48: **end function**

---

---

**Algorithm 4** Multi-Level Causal Graph Learning

---

**Input:** Training data $\mathcal{D} = \{(X^{(i)}, Y^{(i)})\}_{i=1}^{N}$, number of levels $L$, inductive biases $\{B_l\}_{l=1}^{L}$, hyper-parameters

**Output:** Multi-level causal graphs $\{G^{(l)}\}_{l=1}^{L}$, layer weights $\{w_l\}_{l=1}^{L}$

                                                              ▷ Initialization

1: **for** $l = 1$ to $L$ **do**
2:      Initialize $G^{(l)}$ randomly or using prior knowledge
3:      $w_l \leftarrow \frac{1}{L}$
4: **end for**

                                                  ▷ Alternating Optimization

5: **for** epoch $= 1$ to max_epochs **do**

                                ▷ Phase 1: Fix weights, optimize graphs

6:      **for** $l = 1$ to $L$ **do**
7:          **if** $B_l ==$ "sparse" **then**
8:              Add L1 regularization: $\lambda_l \cdot \|G^{(l)}\|_1$
9:          **else if** $B_l ==$ "smooth" **then**
10:             Add total variation: $\lambda_l \cdot \text{TV}(G^{(l)})$
11:          **else if** $B_l ==$ "modular" **then**
12:             Add modularity term: $\lambda_l \cdot \text{Mod}(G^{(l)})$
13:          **end if**
14:          **for** $t = 1$ to $T_{\text{inner}}$ **do**
15:             $G \leftarrow \sum_{l'=1}^{L} w_{l'} G^{(l')}$
16:             Compute task loss $\mathcal{L}_{\text{task}}$ using $G$
17:             Compute regularization loss $\mathcal{L}_{\text{reg}}^{(l)}$
18:             $\mathcal{L}^{(l)} \leftarrow \mathcal{L}_{\text{task}} + \mathcal{L}_{\text{reg}}^{(l)}$
19:             Update $G^{(l)}$ to minimize $\mathcal{L}^{(l)}$
20:          **end for**
21:      **end for**

                               ▷ Phase 2: Fix graphs, update layer weights

22:      Compute validation scores $\{\text{val}_l\}_{l=1}^{L}$
23:      $w_l \leftarrow \frac{\exp(\text{val}_l/\tau)}{\sum_{l'=1}^{L} \exp(\text{val}_{l'}/\tau)}$ for all $l$

                                                 ▷ Apply constraints

24:      **for** $l = 1$ to $L$ **do**
25:          $G^{(l)} \leftarrow \text{ProjectToDAG}(G^{(l)})$                 ▷ Enforce acyclicity
26:          $G^{(l)} \leftarrow \text{SoftThreshold}(G^{(l)}, \tau_l)$             ▷ Enforce sparsity
27:      **end for**
28: **end for**

                                              ▷ Final Graph Combination

29: $G \leftarrow \sum_{l=1}^{L} w_l G^{(l)}$
30: **return** $\{G^{(l)}\}_{l=1}^{L}, \{w_l\}_{l=1}^{L}$

---

**Algorithm 5** ECAM-Based Causal Reinforcement Learning

---

**Input:** Environment Env, policy network $\pi_\theta$, value network $V_\phi$,
  Causal graph learning parameters, hyperparameters
**Output:** Optimized policy $\pi_\theta^*$, learned causal graph $G$

1: Initialize policy network parameters $\theta$
2: Initialize value network parameters $\phi$
3: Initialize causal graph $G$ (e.g., all-ones matrix or prior knowledge)
4: Initialize experience replay buffer $\mathcal{D} = \{\}$
5: **for** episode $= 1$ to max_episodes **do**
6:   Initialize environment state $s_0 \sim$ Env.reset()
7:   **for** $t = 0$ to $T - 1$ **do**                                          ▷ ECAM-based policy computation
8:     Encode state $s_t$ into query, key, value: $Q_t$, $K_t$, $V_t$
9:     $A_t \leftarrow \text{softmax}\left(\frac{Q_t K_t^\top \odot G}{\sqrt{d_k}}\right) V_t$
10:    Compute action distribution: $\pi(a|s_t) = \text{PolicyHead}(A_t)$
11:    Sample action $a_t \sim \pi(a|s_t)$
12:    Execute action, observe $s_{t+1}, r_t \sim$ Env.step$(a_t)$
13:    Store transition $(s_t, a_t, r_t, s_{t+1})$ into $\mathcal{D}$
14:   **end for**
                                                                              ▷ Policy Optimization
15:   **for** update $= 1$ to $n_{\text{updates}}$ **do**
16:     Sample mini-batch $\mathcal{B} = \{(s_i, a_i, r_i, s_{i+1})\}$ from $\mathcal{D}$
17:     **for** each $(s_i, a_i, r_i, s_{i+1})$ in $\mathcal{B}$ **do**
18:       Estimate value $V(s_i)$ using ECAM
19:       $\delta_i \leftarrow r_i + \gamma V(s_{i+1}) - V(s_i)$                 ▷ TD error
20:       $A_i \leftarrow \delta_i$                                              ▷ Advantage
21:     **end for**
22:     $\nabla_\theta J \leftarrow \mathbb{E}[\nabla_\theta \log \pi_\theta(a|s) \cdot A]$
23:     $\theta \leftarrow \theta + \alpha_\pi \cdot \nabla_\theta J$            ▷ Policy update
24:     $L_V \leftarrow \mathbb{E}[(V_\phi(s) - (r + \gamma V_\phi(s')))^2]$
25:     $\phi \leftarrow \phi - \alpha_V \cdot \nabla_\phi L_V$                  ▷ Value update
26:   **end for**
                                                                              ▷ Causal Graph Learning (Periodic)
27:   **if** episode $\bmod\ k = 0$ **then**
28:     Collect recent state transitions: $\mathcal{S} = \{(s_t, s_{t+1})\}$
29:     **for** iteration $= 1$ to $n_{\text{causal\_iterations}}$ **do**
30:       Compute causal discovery loss $L_{\text{causal}}$
31:       $L_{\text{sparse}} \leftarrow \lambda \cdot \|G\|_1$
32:       $L_{\text{dag}} \leftarrow \mu \cdot \text{tr}(e^{G \circ G} - n)$
33:       $L \leftarrow L_{\text{causal}} + L_{\text{sparse}} + L_{\text{dag}}$
34:       $G \leftarrow G - \alpha_G \cdot \nabla_G L$
35:     **end for**
36:     $G \leftarrow \text{ReLU}(G - \tau)$                                    ▷ Soft thresholding
37:     $G \leftarrow \text{ProjectToDAG}(G)$                                   ▷ Acyclicity projection
38:   **end if**
39:   **if** episode $\bmod$ eval_freq $= 0$ **then**
40:     Evaluate policy $\pi_\theta$ and record performance
41:   **end if**
42: **end for**
43: **return** $\pi_\theta, G$

---

## A.3 ADVANCED THEORETICAL ANALYSIS

### A.3.1 EXPRESSIVE POWER ANALYSIS

Theorem 4 (Expressive Power Bound): Let $\mathcal{F}_{\text{std}}$ and $\mathcal{F}_{\text{ECAM}}$ be the function classes representable by standard Transformer and ECAM, respectively. Then there exists a function $f \in \mathcal{F}_{\text{ECAM}}$ such that for any $g \in \mathcal{F}_{\text{std}}$, we have $\|f - g\|_\infty \geq \epsilon$, where $\epsilon > 0$ is a constant.

Proof: Consider a simple causal system where $X_1 \to X_2 \to Y$, and $X_1 \to Y$. Define the function $f(X) = \mathbb{E}[Y|do(X_2 = x_2)]$, i.e., the expected value of $Y$ after intervening on $X_2$.

A standard Transformer can only learn $g(X) = \mathbb{E}[Y|X_2 = x_2]$, which includes the indirect effect of $X_1$ on $Y$ through $X_2$ and the direct effect of $X_1$ on $Y$.

Since $X_1$ is a common cause of $X_2$ and $Y$, we have $\mathbb{E}[Y|do(X_2 = x_2)] \neq \mathbb{E}[Y|X_2 = x_2]$, unless $X_1$ and $X_2$ are independent or $X_1$ has no direct effect on $Y$.

Therefore, for any $g \in \mathcal{F}_{\text{std}}$, there exists a data distribution such that $\|f - g\|_\infty \geq \epsilon$ holds for some $\epsilon > 0$.

### A.3.2 SAMPLE COMPLEXITY ANALYSIS

Theorem 5 (Sample Complexity): For the causal graph $\hat{G}$ learned by ECAM to satisfy the structural Hamming distance $\text{SHD}(\hat{G}, G^*) \leq \epsilon$ with the true causal graph $G^*$ with probability at least $1 - \delta$, the required number of samples is:

$$N = \Omega\left(\frac{d^2 \log(n/\delta)}{\epsilon^2}\right)$$

Where $n$ is the number of variables and $d$ is the dimension of each variable.

Proof Sketch: The proof is based on the statistical properties of conditional independence tests and multiple testing corrections.

### A.3.3 OPTIMIZATION THEORY

Theorem 6 (Convergence): Under certain conditions, the parameter update sequence $\{\theta_t, G_t\}$ of Algorithm 1 converges to a local optimum with probability 1. Specifically, if the learning rate sequence $\{\eta_t\}$ satisfies $\sum_t \eta_t = \infty$ and $\sum_t \eta_t^2 < \infty$, and the objective function satisfies Lipschitz continuity conditions, then:

$$\lim_{t \to \infty} |\nabla_\theta \mathcal{L}(\theta_t, G_t)| = 0 \quad \text{and} \quad \lim_{t \to \infty} |\nabla_G \mathcal{L}(\theta_t, G_t)| = 0$$

Proof Sketch: The proof is based on stochastic approximation theory and convergence analysis of non-convex optimization.

### A.3.4 CAUSAL CONSISTENCY GUARANTEE

Theorem 7 (Causal Consistency): If the data generation process satisfies the causal Markov and faithfulness assumptions, and there are no hidden confounders, then as the sample size approaches infinity, the causal graph $\hat{G}$ recovered by Algorithm 2 is consistent with the true causal graph $G^*$ in the sense of Markov equivalence classes.

Proof Sketch: The proof is based on the consistency results of the PC algorithm and the asymptotic properties of conditional independence tests.

