# OpenReview forum: "ECAM: Enhancing Causal Reasoning in Foundation Models with Endogenous Causal Attention Mechanism"
_ICLR.cc/2026/Conference — ICLR 2026 Conference Withdrawn Submission_

### Official Review · Reviewer_7Biz · 2025-10-30

**Soundness:** 2
**Presentation:** 2
**Contribution:** 2
**Rating:** 4
**Confidence:** 4

**Summary:**

This paper introduces the Endogenous Causal Attention Mechanism (ECAM), a module designed to be integrated into transformer-based architectures. The goal is to address the limitation that standard attention mechanisms capture correlational patterns rather than true causal dependencies. ECAM learns or utilizes a Local Causal Graph (LCG) representing causal relationships between input elements. This graph is then used to modulate the attention scores, for example by masking or re-weighting, to align the attention flow with the causal structure. The authors also propose mechanisms within ECAM to perform intervention-based and counterfactual reasoning. The module is evaluated on causal discovery, causal effect estimation, and downstream tasks including CLUTRR, GLUE, and causal VQA, with reported improved performance.

**Strengths:**

* The core idea is highly intuitive and well-motivated. A standard attention layer can be seen as a fully connected graph where every input element can attend to every other. The paper argues this is purely correlational. Imposing a causal structure, via the LCG, to prune or re-weight these connections is a very logical and principled approach to embed causal reasoning directly into the model's architecture.
* The ablation study is comprehensive and paints a clear picture of the model's performance landscape. It successfully isolates the contributions of the key components, such as LCG learning, the intervention mechanism, and the counterfactual mechanism. The finding that removing LCG learning causes the most significant performance degradation supports the paper's central hypothesis.

**Weaknesses:**

* Performance on Downstream Benchmarks: The claim that this method improves performance on tasks like VQA and GLUE is questionable. The baselines used (e.g., standard BERT/ViT, CATT)  are, by today's standards, very old. I highly doubt that the reported scores are competitive with state-of-the-art models from even several years ago, let alone today.
* Conflict Between Causal Constraints and Task Performance: While the idea is intuitive, imposing causal constraints (like sparsity and acyclicity ) might not actually improve performance on complex perception and language tasks. These constraints may not accurately reflect the true, complex data generation process of natural language. In terms of pure learning, these constraints might act as a form of regularization that hinders the model's ability to learn the very (non-causal) correlations and biases that are often necessary to achieve high scores on these benchmarks. I have personally tried this idea several years ago and found that the reported performance might not be reflective of the status quo.
* Causal Discovery Evaluation Details: The performance boost on causal discovery tasks seems plausible, but the methodology is missing critical details. It is not documented how training and testing were separated. This is crucial, as ECAM appears to be a learning-based method, unlike some baselines. The comparison to NOTEARS  is particularly problematic. NOTEARS is an optimization method that learns the graph "on the fly" from the provided data. ECAM, in contrast, seems to be trained to perform this task, by incorporating NOTEARS-like constraints (e.g., acyclicity ). It would not be surprising if a learning-based method, trained on a distribution of graphs, could achieve better results than a one-shot optimization method. This comparison may not be on equal footing.

**Questions:**

* Regarding Downstream Task Baselines: Could you provide a comparison of ECAM (integrated with a modern backbone) against more contemporary state-of-the-art models on GLUE and VQA? This is necessary to validate the claim that this mechanism provides a practical benefit for general-purpose foundation models.
* Regarding Causal Discovery Experiments: Can you please clarify the exact training and testing protocol for the causal discovery experiments?
* Regarding the NOTEARS Comparison: Given that NOTEARS is an optimization-based discovery algorithm and ECAM appears to be a trained model, how do you justify this direct comparison? Isn't it expected that a model trained on the task distribution would outperform a non-trained method? Could the improvements be attributed to ``meta-learning'' rather than a superior discovery mechanism?
* Regarding LCGs in Language: The notion of a "Local Causal Graph"  for a text sequence seems like a strong simplification. How does the model contend with the inherent ambiguity and context-dependency of relationships in natural language? What evidence do you have that the learned LCGs are capturing true causal structures rather than just sophisticated correlational patterns subject to the acyclicity/sparsity constraints?

---

### Official Review · Reviewer_y8B5 · 2025-10-31

**Soundness:** 2
**Presentation:** 1
**Contribution:** 2
**Rating:** 2
**Confidence:** 2

**Summary:**

The paper introduces the Endogenous Causal Attention Mechanism (ECAM), a novel approach that guides attention heads to follow a causal graph discovered by the model itself. This design enables the model to capture more faithful causal relationships, leading to improved latent representations and enabling causal interventions within the latent space. The author evaluates the proposed method on several datasets, analysing both its ability to recover the true underlying causal graph and its performance on downstream tasks. The results show consistent improvements across multiple datasets.

**Strengths:**

The paper addresses an important open problem, namely, the lack of proper causal structure within attention mechanisms. The experimental setup appears extensive, and the proposed method consistently outperforms competitive baselines. The paper is clearly written and easy to follow, although some crucial details are missing (as discussed in the Weaknesses section).

**Weaknesses:**

I do not believe the paper, in its current form, is ready for acceptance, for the following reasons:
- Crucial methodological details are missing, particularly regarding the loss function used to discover the causal graph and how sparsity and acyclicity are enforced.
- Additional clarification is needed on how interventions and counterfactuals are performed within the attention layers. The corresponding experiments also lack sufficient explanation to fully understand the proposed procedure.
- Figures 2 and 3 are never referenced in the text and do not appear to provide relevant insights for the paper. Moreover, they include results from unrelated datasets and methods, and the plots themselves seem incomplete.

**Questions:**

See weaknesses

---

### Official Review · Reviewer_52X3 · 2025-10-31

**Soundness:** 2
**Presentation:** 2
**Contribution:** 1
**Rating:** 2
**Confidence:** 4

**Summary:**

The paper proposes ECAM: a method integrating causal structure directly into the attention mechanism of transformers. The method considers the input text as a set of causal variables and builds a causal graph from it. The causal graph is then applied as a mask over the attention matrix to induce causal structure into the reasoning process of the transformer. Experiments show an increased performance on causal reasoning tasks compared to the presented baselines.

**Strengths:**

1. The paper tackles a very challenging problem in causal inference with LLMs with an original and interesting approach. Some of the weaknesses below can be traced back to the complexity of the causal inference problem when compared to the simplicity of the proposed approach rather than the soundness of the proposed method.
2. The paper is easy to follow and the methods are presented clearly.

**Weaknesses:**

1. The main issue of the method is the confusion between the system of causal variables $X$ and the input tokens. The method seems to operate on tokens while considering them as causal variables although no part of the method extracts high-level causal representations. This has a significant impact on the understanding of the method: interventions on tokens can easily be performed with existing methods but do not correspond to causal interventions on semantic concepts.
2. The paper does not explain how the method works at different layers. As the input embeddings aggregate information from previous layers, input vectors do not necessarily have clear understandable semantics, limiting the interpretability of the causal relationships or the interventions.
3. Some of the references are incorrect. For instance, [Geiger et al., 2023] in the first paragraph of Section 3.2 instead refers to the paper "Causal interpretation of self-attention in pre-trained transformers" [Rokehar, Gurwicz and Nisimov, 2023].
4. In addition to the standard task loss, the only added training signal is a sparsity and acyclicity regularization term. While the authors mention that causal discovery methods like the PC algorithm could be integrated, from my understanding, the proposed method does not include any causal discovery tests. The resulting dependency graph obtained cannot be considered to be causal.
5. While the method allow for interventions, the way counterfactuals are computed is unclear. the authors mention performing two forward passes (with an without interventions) and "computing the counterfactual attention output for $x_j$.", this is very vague and hard to understand. What does this $x_j$ correspond to?
6. Theorem 1 is very vague and not justified: conditions under which ECAM can distinguish additional graph structures than attention are not specified and I could not find the proof for the theorem. the same applies to Theorems 2 and 3. In particular, the assumption of correctness of the causal discovery method justifying Theorem 3 is not justified.
7. Adding experiments with more recent LLM baselines would be greatly beneficial to the paper as LLMs have demonstrated great in-domain causal discovery abilities while being limited in out-of-distribution settings [1,2,3].
8. It is unclear from the setup description how the experiments are performed. In particular, transformers do not expect the same input types as standard causal discovery algorithms like PC, GES, etc. Explaining how inputs are provided to the models and outputs gathered would be helpful for the reader.
9. As a follow-up comment, the training procedure and hyperparameters are not described.
10. Result figures are hard to interpret. Figure 2 does not provide a scale for the plots. Moreover, ECAM performs much worse than other baselines if the metric is RMSE as described in the legend. Similarly, Figure 3 does not specify what causal inference tasks are presented in the plots.

[1] Jin, Z., Liu, J., Lyu, Z., Poff, S., Sachan, M., Mihalcea, R., ... & Schölkopf, B. (2023). Can large language models infer causation from correlation?. arXiv preprint arXiv:2306.05836.

[2] Zečević, M., Willig, M., Dhami, D. S., & Kersting, K. (2023). Causal parrots: Large language models may talk causality but are not causal. arXiv preprint arXiv:2308.13067.

[3] Jin, Z., Chen, Y., Leeb, F., Gresele, L., Kamal, O., Lyu, Z., ... & Schölkopf, B. (2023). Cladder: Assessing causal reasoning in language models. Advances in Neural Information Processing Systems, 36, 31038-31065.

**Questions:**

1. In Figure 1, conditional independence tests are mentioned as part of the method but are not discussed later in the paper. What are the testing methods used and what training data is used for the tests?
2. In the description of the way counterfactuals are computed, what does this $x_j$ correspond to?
3 Can you provide the proofs of theorems 1 and 2?
4. What is the computational overhead induced by ECAM compared to standard attention mechanisms?
5. How are the inputs provided to the transformers compared to the standard causal structure discovery algorithms? How are the outputs extracted? How did you ensure that the results were comparable between models?

---

### Official Review · Reviewer_WeK4 · 2025-11-01

**Soundness:** 3
**Presentation:** 2
**Contribution:** 2
**Rating:** 4
**Confidence:** 4

**Summary:**

This paper proposes a novel causal attention mechanism called ECAM. Unlike previous works that use causal graphs merely as post-hoc filters or constraints, ECAM directly embeds the causal graph into the attention computation. It can generate different causal graphs for different data, significantly enhancing generalizability and flexibility. Moreover, ECAM supports both intervention-based and counterfactual-based attention computation, strengthening its capacity for causal reasoning.

**Strengths:**

- A causal-graph-embedded attention mechanism is proposed, which significantly enhances the generalizability of causal attention and helps prevent foundation models from overly relying on spurious correlations.
- The paper accurately identifies the shortcomings of previous methods and addresses them by enabling ECAM to support both intervention-based and counterfactual attention computation.

**Weaknesses:**

- The paper provides very unclear descriptions of the key components.
- The statements of the theorems are vague, and some may not even qualify as proper theorems. (Details see Questions.)

**Questions:**

- The allocation of space in the paper is not well balanced, as many key components are not described in detail. For example, what exactly is the _local causal graph_? In Figure 2, how does the _graph structure learning_ module work specifically? What method is used for the _conditional independence test_—is it the PC algorithm or something else? I am familiar with works like NoTears, but for readers without a background in causal learning, it would be difficult to understand.
- Theorem 1 seems more like an observation based on experimental results. Additionally, what exactly does the term “certain condition” refer to?

---

### Note · Authors · 2025-11-19

I have read and agree with the venue's withdrawal policy on behalf of myself and my co-authors.